# Initial Age and Performans Status: Predicators for Re-Irradiation Ability in Patients with Relapsed Brain Metastasis after Initial Stereotactic Radiotherapy

**DOI:** 10.3390/cancers16142602

**Published:** 2024-07-21

**Authors:** Isabelle Chambrelant, Laure Kuntz, Clara Le Fèvre, Delphine Jarnet, Julian Jacob, Georges Noël

**Affiliations:** 1Department of Radiation Oncology, Institut de Cancérologie Strasbourg Europe (ICANS), UNICANCER, Paul Strauss Comprehensive Cancer Center, 67200 Strasbourg, France; i.chambrelant@icans.eu (I.C.); l.kuntz@icans.eu (L.K.); c.lefevre@icans.eu (C.L.F.); 2Department of Medical Physics, Institut de Cancérologie Strasbourg Europe (ICANS), UNICANCER, Paul Strauss Comprehensive Cancer Center, 67200 Strasbourg, France; d.jarnet@icans.eu; 3Department of Radiation Oncology, AP-HP, Sorbonne Université, Hôpitaux Universitaires Pitié-Salpêtrière, 47-83 Bd de l’Hôpital, CEDEX 13, 75651 Paris, France; julian.jacob@aphp.fr

**Keywords:** single brain metastases, stereotactic radiation therapy, acute toxicities, local recurrence, overall survival

## Abstract

**Simple Summary:**

Brain metastases (BMs) are common in cancer patients, and stereotactic radiation therapy (SRT) is a preferred treatment. This retrospective study analyzed patients treated with SRT for a single BM and compared two subgroups: “Cohort 1” (no cerebral re-irradiation) and “Cohort 2” (received subsequent SRT sessions for recurrence). Patients treated with SRT for a single BM between January 2010 and June 2020 were included. Cohort 1 had 152 patients, and Cohort 2 had 46 patients. Cohort 2 had younger patients with higher Karnofsky performance status (KPS). Median overall survival was longer in Cohort 2 (21.8 months) compared to Cohort 1 (6.1 months). Recurrence rates were higher in Cohort 2 (*p* < 0.001), likely due to patient selection and longer survival. Age and KPS were predictive of survival, especially for patients under 65 with KPS > 80. Age and KPS predict better survival in patients with BMs.

**Abstract:**

Background: Brain metastases (BMs) frequently occur in cancer patients, and stereotactic radiation therapy (SRT) is a preferred treatment option. In this retrospective study, we analyzed patients treated by SRT for a single BM during their first SRT session and we compared two subgroups: “Cohort 1” with patients did not undergo cerebral re-irradiation and “Cohort 2” with patients received at least one subsequent SRT session for cerebral recurrence. Methods: We included patients who received SRT for a single BM between January 2010 and June 2020. Cohort 1 comprised 152 patients, and Cohort 2 had 46 patients. Results: Cohort 2 exhibited younger patients with higher Karnofsky performance status (KPS). Median overall survival was considerably longer in Cohort 2 (21.8 months) compared to Cohort 1 (6.1 months). Local and cerebral recurrence rates were significantly higher in Cohort 2 (*p* < 0.001), attributed to patient selection and longer survival. The combined score of age and KPS proved to be a predictive factor for survival, with patients under 65 years of age and KPS > 80 showing the best survival rates in the overall population. Conclusion: This retrospective study highlights that the combined score of age and KPS can predict better survival, especially for patients under 65 years with a KPS score above 80. Further research involving larger and more diverse populations is essential to validate and expand upon these findings.

## 1. Introduction

Brain metastases (BMs) are the most common intracranial tumors and occur in 20 to 40% of cancer patients [1]. Regardless of the histological type of the primary tumor, about 70% of patients typically have a solitary BM [2]. Stereotactic radiation therapy (SRT) is an advanced and accurate treatment method that utilizes converging microbeams to administer concentrated and high radiation doses to extremely small areas. SRT has become a major approach for treating BMs and has largely supplanted whole-brain radiation therapy (WBRT) [3,4].

Despite the availability of treatment options such as SRT, WBRT, surgical resection, or systemic therapy, patients have a poor prognosis, with a median survival period ranging from 3 to 47 months [5,6]. The issue of BMs is multifaceted due to the significant variability among patients in this population. Various prognostic indices have been developed to guide physicians to choose the better treatment, such as the Diagnosis-Specific Graded Prognostic Assessment (DS-GPA) score [6], as well as the Recursive Partitioning Analysis (RPA) classification [7,8]. However, even within the same subgroup, patient survival can vary significantly [6,7,8].

The presence of a solitary BM at diagnosis is widely acknowledged as a favorable prognostic factor [9], and the count of BMs is a component integrated into several predictive scoring systems [6,10]. Nevertheless, in our center, we noticed a discrepancy in survival outcomes among patients who initially received SRT for a single BM. Specifically, some patients survived for an extended duration, enabling them to potentially undergo a subsequent cerebral irradiation if a local or regional cerebral recurrence occurred, whereas others did not have the opportunity for a second cerebral irradiation [11].

The aim of this study was to retrospectively analyze data of patients who were treated with SRT for a single BM at the first irradiation and compare characteristics and follow-up outcomes of two subgroups, including one group in which patients were re-irradiated for consecutive BMs, while the other was not. The expected outcome of this analysis was to determine whether patient characteristics differed between the two subgroups, which could explain the possibility of re-irradiation for some patients and not for others.

## 2. Materials and Methods

### 2.1. Patient and Treatment Modalities

We used our institutional database to obtain the list of patients who received SRT for BMs treatment between January 2010 and June 2020. We excluded patients who had multiple BMs at the first session, brainstem metastases, resected BMs, and WBRT before or after SRT. Among 1240 patients treated with stereotactic irradiation for BMs, we identified 198 patients treated for a single BM at the first irradiation. We decided to analyze data in two subgroups: “Cohort 1” treated for a single BM without cerebral re-irradiation (152 patients, 76.8%) and “Cohort 2” treated for a single BM with at least one subsequent brain stereotactic irradiation sequence for a local or regional cerebral recurrence (46 patients, 23.2%).

Patients received one fraction of 14 Gy (e.g., at 2 mm of the metastases margin (PTV) and with a 20 Gy dose at isocenter) (n = 41; 20.7%) or 23.1 Gy (e.g., at 2 mm of the metastases margin (PTV) and with a 33 Gy dose at isocenter) in three fractions every other day (n = 157; 79.3%), if the size of the BM was smaller or greater than 1 cm, respectively. Treatments were delivered using dynamic conformal arc therapy (DCAT) or volumetric modulated arc therapy (VMAT). Treatments were planned on an iPlan^®^ RT Dose V4.5.5 (Brainlab^®^ AG, Feldkirchen, Germany) or on an Eclipse^®^ System V15.6 (Varian Medical Systems^®^, Palo Alto, CA, USA) with a 1.25 mm grid calculation. Treatments were delivered by an SRS-6MV beam from a Novalis Tx™ (Varian Medical Systems^®^, Palo Alto, CA, USA) or by a 6MV-FFF beam from a TrueBeam STx™ (Varian Medical Systems^®^, Palo Alto, CA, USA).

For each patient, we collected age, KPS, primitive cancer, delivery and type of systemic treatment, extracerebral disease, BM’s localization, and size. We calculated the RPA and DS-GPA scores [6,7,8].

### 2.2. Acute Toxicities Reporting

Each patient underwent an initial consultation with their referring radiation oncologist to strategize SRT. During this consultation, details regarding medical history, prior oncological treatments, corticosteroid therapy, and various neurological symptoms were documented. Throughout the course of SRT, patients had weekly consultations with the radiation oncologist to assess treatment tolerance. The administration of corticosteroid therapy and the presence of neurological symptoms such as confusion, headaches, nausea and vomiting, epilepsy, sensory-motor disorders, and dizziness were recorded and graded using the Common Terminology Criteria for Adverse Events version 5.0 (CTCAE v.5.0) [12].

### 2.3. Follow-Up

Magnetic resonance imaging (MRI) was performed every three months in the first two years after SRT then every six months to evaluate therapeutic efficacy and diagnose local recurrence (LR), or cerebral recurrence (CR), and potential radionecrosis (RN). A newly observed contrast enhancement outside the previously treated BM was classified as CR. Contrast enhancement within the previously treated BM indicated either LR or RN. To distinguish between these possibilities, patients underwent 18-fluorodeoxyglucose (FDG) PET-CT, surgical intervention, corticosteroid testing, or a repeat MRI within a short timeframe [13,14,15,16].

### 2.4. Statistical Analysis

#### Group Comparison

Numeric variables were expressed as mean (±SD) and discrete outcomes as absolute and relative (%) frequencies. We created 2 groups according to the values of the cohort. Group comparability was assessed by comparing baseline demographic data and follow-up duration between groups. Normality and heteroskedasticity of continuous data were assessed with Shapiro–Wilk and Levene’s test, respectively. Continuous outcomes were compared with unpaired Student *t*-test, Welch *t*-test, or Mann–Whitney U test according to data distribution. Discrete outcomes were compared with chi-squared or Fisher’s exact test accordingly. The alpha risk was set to 5% and two-tailed tests were used.

We used the Kaplan–Meier method to estimate survival probabilities from “Date of SRT” until “Date of Death” and their pointwise 95% confidence intervals. The Logrank non-parametric test for comparison of survival distributions was used to compare survival differences between groups. The alpha risk was set to 5.0%.

Statistical analysis was performed with EasyMedStat (version 3.27; www.easymedstat.com, accessed on 08/15/2023).

The study follows the French laws mandatorily required by the CNIL (Commission Nationale de I’informatique et des Libertés) and was declared to this French institution using the MR004 form on https://www.health-data-hub.fr/ (n° F20201119113809, accessed on 11/23/2020) and this study was accepted by the Institution Review Board of ICANS (Cancer Institute of Strasbourg-Europe) (n° IRB-2023-1).

## 3. Results

### 3.1. Patients’ Characteristics

Treatments were performed between January 2010 and June 2020. Overall, the median age at treatment was 66.5 years old (range 32–91 years old). Patients in Cohort 1 were significantly older than patients in Cohort 2 (mean age: 67 versus 63.3 years old, respectively, *p* = 0.031). Patients in Cohort 2 presented a significantly greater KPS, 65.2% of patients had a superior KPS at 80% in Cohort 2 compared to 34.9% in Cohort 1 (*p* < 0.001) (Appendix A). Distribution of patients according to their RPA group and DS-GPA class was not significantly different between the two cohorts. The mean time between primitive cancer diagnosis and BM treatment was 36.4 months (range 0–288 months) in Cohort 1 and 33.1 months (range 0–264 months) in Cohort 2 (*p* = 0.391). There was no significant difference between the groups for nature of primitive cancer, initial TNM, presence of extracranial metastases, and control of the primary tumor. Table 1 summarize the patients’ characteristics and their comparisons between the two cohorts.

### 3.2. Brain Metastases and Treatment Characteristics

The mean volume of BMs was 5.7 cc (range 0.03–61.8 cc) in Cohort 1 and 5.8 cc (range 0.03–39.6 cc) in Cohort 2 (*p* = 0.93). The localization of BMs (supra- or sub-tentorial) was not significantly different between cohorts (*p* = 0.818). There was no significant difference between the groups for technique of SRT used and prescribed dose. Most patients, 146 (96%) and 46 (100%), received corticosteroids during irradiation in Cohort 1 and 2, respectively, (*p* = 0.339), followed by a progressive decrease. Table 2 summarize brain metastases and treatments’ characteristics and their comparisons between the two cohorts.

### 3.3. Acute Toxicities and Follow-Up

Acute toxicities are detailed in Table 3. No patient had grade 3 and 4 neurologic symptoms before or during the treatment. The most frequent grade 1 and 2 symptoms were headache and sensory-motor deficits, but these were most often already present before treatment and their grade did not change. There was not a significant difference between cohorts concerning acute toxicities of SRT (*p* = 0.497) and the dose of corticosteroids was increased in 11 (7.3%) patients in Cohort 1 and one (2.2%) patient in Cohort 2 during treatment (*p* = 0.302).

### 3.4. Follow-Up Outcomes

Table 4 summarizes follow-up outcomes. The mean follow-up after SRT was 16.8 months (range 0–140.7 months) in Cohort 1 and 30.8 months (range 2–101 months) in Cohort 2 (*p* < 0.001).

#### 3.4.1. Radionecrosis

Thirteen patients (6.6%) developed radionecrosis (RN) considering both groups. Incidence of RN was not significantly different between the cohorts: eight (5.3%) patients in Cohort 1 and five (10.9%) patients in Cohort 2 (*p* = 0.185). The mean time between SRT and diagnosis of RN was 12.4 months and 10.6 months in Cohort 1 and Cohort 2, respectively (*p* = 0.55).

#### 3.4.2. Local Recurrence

Incidence of local recurrence (LR) was significantly higher in Cohort 2 than in Cohort 1 (28.3% in Cohort 2 vs. 5.3% in Cohort 1, *p* < 0.001). The mean time between SRT and LR was 9.4 months and 13.5 months in Cohort 1 and Cohort 2, respectively (*p* = 0.36). The 3-, 6-, and 12-month local control (LC) rates were 99.1%, 97.9%, and 94.9%, respectively, for Cohort 1 and 97.8%, 90.9%, and 80.5%, respectively for Cohort 2. There was no significant difference for 3-, 6-, and 12-month LC rates between groups (*p* = 0.61, 0.113 and 0.09, respectively).

#### 3.4.3. Cerebral Recurrence

Incidence of cerebral recurrence (CR) was significantly higher in Cohort 2 than in Cohort 1 (82.6% in Cohort 2 vs. 9.2% in Cohort 1, *p* < 0.001). The mean time between SRT and CR was 9.2 months and 11.4 months in Cohort 1 and Cohort 2, respectively (*p* = 0.448). The 3-, 6-, and 12-month cerebral progression-free survival (c-PFS) rates were 96.7%, 92.7%, and 88.1%, respectively, for Cohort 1 and 78.3%, 50%, and 39.1%, respectively for Cohort 2. There was a significant difference between groups for 3-, 6-, and 12-month c-PFS rates (*p* = 0.003, <0.0001, and <0.0001, respectively).

#### 3.4.4. Overall Survival

Patients died in 86.8% and 89.1% in Cohort 1 and 2, respectively (*p* = 0.804). However, the median overall survival (OS) was 6.1 months in Cohort 1 and 21.8 months in Cohort 2. There was a difference between survival distributions of both cohorts (*p* = 0.003). The 12-month OS rates were 30.3% (95% CI: 23.2–37.7) for Cohort 1 and 78.3% (95% CI: 63.4–87.7) for Cohort 2 and the 24-month OS rates were 20.4% (95% CI: 14.4–27.1) for Cohort 1 and 43.5% (95% CI: 29.0–57.1) for Cohort 2 (Figure 1).

As described above, the only variables significantly different between Cohort 1 and 2 were age and KPS. We therefore used a combined score of these two variables. The best option was age < 65 and KPS > 80, and the worst option was age ≥ 65 and KPS ≤ 80. We compared OS in these two groups (Figure 2). There was a significant difference between survival distributions of “age < 65 and KPS > 80” and “age ≥ 65 and KPS ≤ 80” (*p* = 0.0006). The 6-month OS rates were 77.8% (95% CI: 62.6–87.4) and 44.6% (95% CI: 33.1–55.5), respectively, and the 12-month rates were 57.8% (95% CI: 42.1–70.6) and 24.3% (95% CI: 15.3–34.5), respectively.

## 4. Discussion

Approximately a third of cancer patients develops BMs during their illness, causing a significantly poor oncological prognosis. This incidence is likely to increase with the development of new therapies such as targeted therapies and immunotherapy, which prolong patients’ lives. Despite advances in molecule distribution, systemic therapies often struggle to reach the brain, resulting in frequent instances of relapse [17]. Indeed, approximately 50 to 70% of patients who had received initial SRT developed new BM within one year after the first SRT [18,19]. This retrospective study tried to understand why some patients initially treated for a single BM had a longer survival, allowing them to be eligible for a second SRT session while others could not reach a second session of SRT. The single-center design of this study guarantees a level of uniformity in prescribing, treatment delivery and clinical or radiological follow-up. A substantial amount of data were gathered, with minimal missing information.

In our study, we confirmed that SRT is a safe treatment. Indeed, we observed that none of the patients experienced grade 3 or 4 acute toxicity. We also documented the neurological symptoms before and after irradiation. Notably, almost all the neurological symptoms reported by the patients before irradiation remained stable throughout the SRT session. Our findings regarding acute toxicity align closely with the study conducted by Jimenez et al. [20]. In their study, they investigated the acute toxicity of SRT in 156 patients treated for a single BM. They reported that around 24% of patients encountered at least one adverse symptom potentially related to SRT, with fatigue and headache being the most frequently observed symptoms.

In terms of late toxicities, our study revealed that 13 patients (6.6%) developed RN in the overall population. This incidence was lower than what has been reported in the literature [21,22]. However, the median time of both cumulated cohorts, between SRT and RN (9 months), was closer than these two studies (10.7 and 11 months) [21,22]. The lower incidence of RN in our study can be attributed, in part, to the fact that no patients in the current study had prior WBRT, which has the potential to increase RN rates by improving the cumulative biologically effective dose [23]. Additionally, our median follow-up duration is shorter than that of the compared studies [21,22] and is equal to our median time between SRT and RN. Consequently, it is possible that some cases of RN were underdiagnosed in our study due to the early mortality of our patients compared to the reported literature [21,22].

LC results of the current study were found to be comparable to the findings of recent studies that also investigated the use of SRT for single BM [24,25,26]. Regarding the comparison between Cohort 1 and Cohort 2, we observed significantly higher rates of LR and CR in Cohort 2 than in Cohort 1. This could be explained by patient selection bias. Indeed, the patients in Cohort 2 were systematically re-irradiated, so they inevitably recurred locally or in the brain. Moreover, patients in Cohort 2 lived longer and were therefore more likely to relapse. Finally, follow-up was longer in Cohort 2 compared to Cohort 1, and the longer the follow-up period, the higher the CR and LR rates. However, it is crucial to highlight that the mean time to LR or CR is approximately equal to the mean time to death in Cohort 1 (9.4 months, 9.2 months, and 9 months), so many patients died before there was time for recurrence. On the contrary, the mean time to death in Cohort 2 is approximately twice the mean time to LR or CR (24.8 months, 13.5 months, and 11.4 months) giving patients time to relapse.

In Cohort 1, the follow-up outcomes, particularly the median OS, were found to be inferior to those reported in the existing literature. Specifically, we observed a median OS of only 6.1 months, while previous studies by other authors have documented a median OS ranging from 10 to 15 months [27,28,29]. This discrepancy could be attributed to our stringent inclusion criteria, particularly the exclusion of cerebral re-irradiation in Cohort 1. We have previously shown that patients in this cohort were not re-irradiated, as the mean OS was equal to the mean time to CR. On the other hand, due to the same reasons of patient selection bias, patients in Cohort 2 exhibited a median OS higher to the literature, with 21.8 months.

To comprehend the substantial difference in OS between the two cohorts, we conducted an analysis of survival within two distinct groups formed by a combined score comprising the two criteria who significantly differed between the cohorts: age and KPS. The results revealed a significantly higher OS for the “age < 65 and KPS > 80” group in comparison to the “age ≥ 65 and KPS ≤ 80” group. Thus, this combined score appears to be predictive of survival in this study population, and consequently younger patients with the best KPS are at greater risk of re-irradiation. It is therefore crucial to use an SRT technique that minimizes doses to healthy tissue in this group [30]. Furthermore, a recent study has demonstrated that these two criteria remain stable throughout the process of re-irradiation [31]. Consequently, it is probable that the decision to re-irradiate patients is driven by their life expectancy, which is influenced by age and KPS, rather than re-irradiation itself leading to extended survival. Hence, it is advisable to accord higher priority to these two variables when making re-irradiation decisions. This is in addition to the established RPA and DS-GPA scores, which were not initially designed for re-irradiation purposes [6,7,8]. Becco Neto et al. similarly reported that a KPS score greater than 80 was linked to more favorable survival rates after 10 years in their study on untreated single non-small cell lung cancer BMs. The fact that age did not emerge as a predictive factor of survival in their cohort could be linked to a cutoff at 70 years old that was used for their analysis [32].

The current study adopted a retrospective approach, which could introduce certain limitations in interpreting the findings. Retrospective studies are prone to biases related to selection, reporting, and recall, which might impact the conclusions drawn. Specifically, the cause of death is an unattainable variable within the current study, as it is infrequently documented in medical records and sometimes death was the result of intricate causes. These data would be valuable in shedding light on the reasons for the swifter demise of patients in Cohort 1. Furthermore, because of the strict inclusion criteria, the number of patients remained limited and may compromise the generalizability of our results to a broader population. Nonetheless, this approach allowed us to concentrate on a homogeneous group, facilitating a detailed assessment of the efficacy and tolerability of SRT in these specific patients.

## 5. Conclusions

This retrospective series of 196 patients, treated for a single BM at first SRT, aimed to explore factors influencing survival and the risk of re-irradiation. All patients included in this study originated from a single institution, ensuring consistency in treatment protocols and follow-up procedures. Our findings suggest that the combined score, based on age and KPS, can serve as a predictive factor for improved survival in our study population, especially for patients below 65 years of age with a KPS score of 80 or higher. Overall, our study contributes to increasing the body of evidence supporting the safety and efficacy of SRT in treating BM. The results underscore the significance of patient selection and precise dosimetry to enhance treatment outcomes. Further research involving larger and more diverse cohorts will be essential to validate and expand upon our observations.

## Figures and Tables

**Figure 1 cancers-16-02602-f001:**
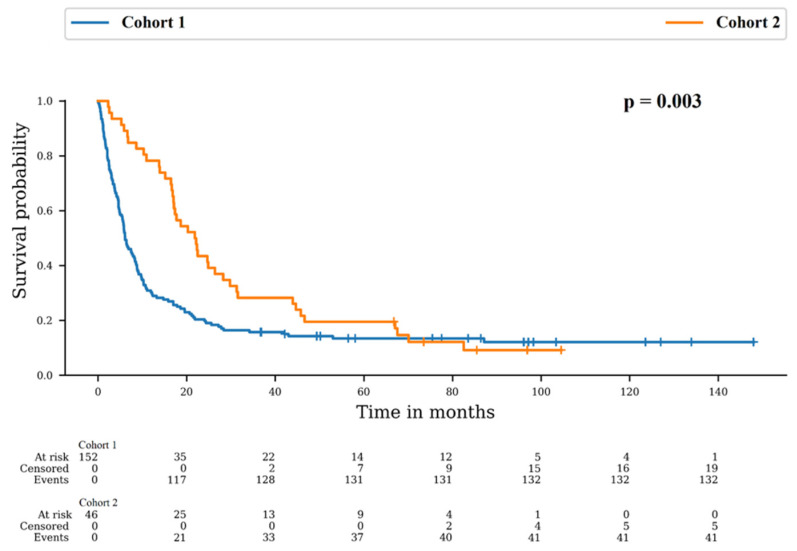
Survival probability in Cohort 1: no re-irradiation and Cohort 2: at least one more irradiation course.

**Figure 2 cancers-16-02602-f002:**
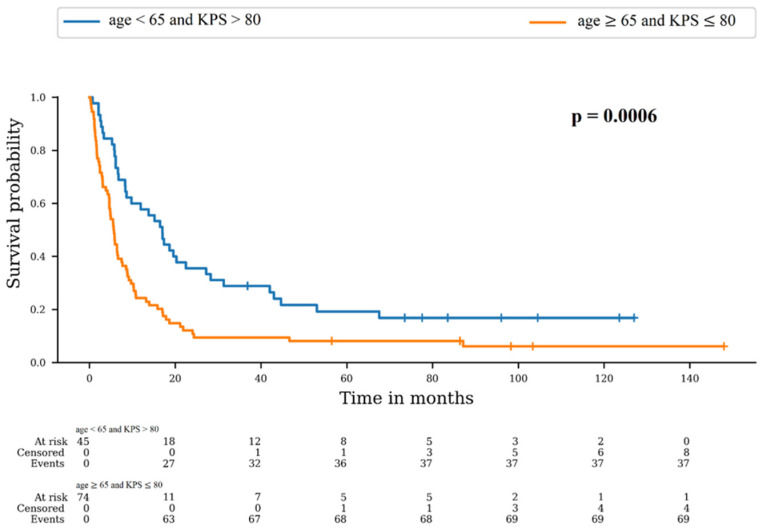
Survival probability according to combined score of age and KPS.

**Table 1 cancers-16-02602-t001:** Patients’ characteristics and their comparisons between the two cohorts. Cohort 1: no re-irradiation and Cohort 2: at least one more irradiation course.

Variables		Global Population (*n* = 198)	Cohort 1 (*n* = 152)	Cohort 2 (*n* = 46)	*p*-Value
Gender					>0.999
	Male	103 (52.0%)	79 (52.0%)	24 (52.1%)	
	Female	95 (48.0%)	73 (48.0%)	22 (47.9%)	
Age at treatment (years)					0.031
	Mean	66 [32–91]	67 [32–89]	63.3 [32–91]	
	Median	66.5	67.5	62	
Time between primitive diagnosis and BM treatment (months)					0.391
	Mean	35.6 [0–288]	36.4 [0–288]	33.1 [0–264]	
	Median	17	17.5	12	
Nb of patients with BM at the diagnosis		39 (19.7%)	29 (19.1%)	10 (21.7%)	0.677
KPS					<0.001
	>80%	83 (41.9%)	53 (34.9%)	30 (65.2%)	
	≤80%	115 (58.1%)	99 (65.1%)	16 (34.8%)	
RPA group					0.486
	I	17 (8.6%)	11 (7.3%)	6 (13.0%)	
	IIa	53 (26.8%)	40 (26.3%)	13 (28.3%)	
	IIb	66 (33.3%)	49 (32.2%)	17 (37.0%)	
	IIc	42 (21.2%)	35 (23.0%)	7 (15.2%)	
	III	20 (10.1%)	17 (11.2%)	3 (6.5%)	
DS-GPA class					0.406
	1	38 (19.2%)	32 (21.0%)	6 (13.0%)	
	2	83 (42.0%)	64 (42.1%)	19 (41.3%)	
	3	57 (28.8%)	41 (27.0%)	16 (34.8%)	
	4	10 (5.0%)	9 (5.9%)	1 (2.2%)	
	NA	10 (5.0%)	6 (4.0%)	4 (8.7%)	
Primitive cancer					0.121
Lungs		119 (60.1%)	87 (57.2%)	32 (69.5%)	
	Adenocarcinoma	70 (58.8%)	49 (56.3%)	21 (65.6%)	
	Epidermoid	32 (26.9%)	27 (31.0%)	5 (15.6%)	
	Small cells	17 (14.3%)	11 (12.7%)	6 (18.8%)	
Breast		23 (11.6%)	20 (13.1%)	3 (6.5%)	
	RH+ HER2-	18 (78.3%)	16 (80.0%)	2 (66.7%)	
	RH+ HER2+	2 (8.7%)	1 (5.0%)	1 (32.3%)	
	RH- HER2+	3 (13.0%)	3 (15.0%)	0	
Digestive		18 (9.1%)	17 (11.2%)	1 (2.2%)	
Melanoma		17 (8.6%)	12 (7.9%)	5 (10.9%)	
	BRAF mutation	9 (52.9%)	7 (58.3%)	2 (40.0%)	
Kidney		11 (5.6%)	10 (6.6%)	1 (2.2%)	
Others		10 (5.0%)	6 (4.0%)	4 (8.7%)	
Initial tumor stage					0.589
	1	21 (10.6%)	17 (11.2%)	4 (8.7%)	
	2	48 (24.3%)	34 (22.4%)	14 (30.4%)	
	3	46 (23.2%)	35 (23.0%)	11 (23.9%)	
	4	45 (22.7%)	37 (24.3%)	8 (17.4%)	
	NA	38 (19.2%)	29 (19.1%)	9 (19.6%)	
Initial node stage					0.472
	0	65 (32.8%)	52 (34.2%)	13 (28.2%)	
	1	29 (14.7%)	24 (15.8%)	5 (10.9%)	
	2	39 (19.7%)	30 (19.7%)	9 (19.6%)	
	3	27 (13.6%)	18 (11.9%)	9 (19.6%)	
	NA	38 (19.2%)	28 (18.4%)	10 (21.7%)	
Initial metastases stage					0.397
	0	108 (54.5%)	80 (52.6%)	28 (60.9%)	
	1	77 (38.9%)	62 (40.8%)	15 (32.6%)	
	NA	13 (6.6%)	10 (6.6%)	3 (6.5%)	
Extracranial metastases					0.689
	Yes	122 (61.6%)	92 (60.5%)	30 (65.2%)	
	No	76 (38.4%)	60 (39.5%)	16 (34.8%)	
Control of the primary tumor site					0.332
	Yes	72 (36.4%)	52 (34.2%)	20 (43.5%)	
	No	126 (63.6%)	100 (65.8%)	26 (56.5%)	

**Table 2 cancers-16-02602-t002:** Brain metastases and treatments’ characteristics and their comparisons between the two cohorts. Cohort 1: no re-irradiation and Cohort 2: at least one more irradiation course.

Variables		Global Population (*n* = 198)	Cohort 1 (*n* = 152)	Cohort 2 (*n* = 46)	*p*-Value
Size (mm)					0.491
	Mean	17 [3–57]	16.8 [3–57]	17.8 [4–49]	
	Median	15	15	16.5	
Volume (cc)					0.930
	Mean	5.7 [0.03–61.8]	5.7 [0.03–61.8]	5.8 [0.1–39.6]	
	Median	2.6	2.5	2.5	
Localization					0.818
	Supra-tentorial	151 (76.3%)	117 (77.0%)	34 (73.9%)	
	Sub-tentorial	47 (23.7%)	35 (23.0%)	12 (26.1%)	
Technique of treatment					0.098
	DCAT	168 (84.8%)	125 (82.2%)	43 (93.5%)	
	VMAT	30 (15.2%)	27 (17.8%)	3 (6.5%)	
Prescribed dose at isocenter					0.829
	33 Gy	157 (79.3%)	120 (79.0%)	37 (80.4%)	
	20 Gy	41 (20.7%)	32 (21.0%)	9 (19.6%)	
Associated treatment					0.339
	Corticosteroids	192 (97.0%)	146 (96.0%)	46 (100%)	

**Table 3 cancers-16-02602-t003:** Acute toxicities and their comparisons between the two cohorts. Cohort 1: no re-irradiation and Cohort 2: at least one more irradiation course.

Variables		Global Population (*n* = 198)	Cohort 1 (*n* = 152)	Cohort 2 (*n* = 46)	*p*-Value
Symptom before treatment					0.030
	Grade 0	110 (55.6%)	91 (59.9%)	19 (41.3%)	
	Grade 1	67 (33.8%)	49 (32.2%)	18 (39.1%)	
	Grade 2	21 (10.6%)	12 (7.9%)	9 (19.6%)	
Symptom during treatment					0.497
	Grade 0	109 (55.0%)	87 (57.2%)	22 (47.9%)	
	Grade 1	69 (34.9%)	51 (33.6%)	18 (39.1%)	
	Grade 2	20 (10.1%)	14 (9.2%)	6 (13.0%)	
Increased corticosteroids during RT		12 (6.1%)	11 (7.3%)	1 (2.2%)	0.302

**Table 4 cancers-16-02602-t004:** Follow-up outcomes and their comparisons between the two cohorts. Cohort 1: no re-irradiation and Cohort 2: at least one more irradiation course.

Variables		Global Population (*n* = 198)	Cohort 1 (*n* = 152)	Cohort 2 (*n* = 46)	*p*-Value
Follow-up post-SRT (months)					<0.001
	Mean	20.1 [0–141]	16.8 [0–140.7]	30.8 [2–101]	
	Median	9	6.1	22	
Radionecrosis		13 (6.6%)	8 (5.3%)	5 (10.9%)	0.185
Delay between SRT and radionecrosis (months)					0.550
	Mean	11.7 [3–45]	12.4 [3–45]	10.6 [5–15]	
	Median	9	7.5	12	
Local recurrence		21 (10.6%)	8 (5.3%)	13 (28.3%)	<0.001
Delay between SRT and local recurrence (months)					0.360
	Mean	11.9 [2–40]	9.4 [3–24]	13.5 [2–40]	
	Median	9	10.5	11	
Cerebral recurrence		52 (26.3%)	14 (9.2%)	38 (82.6%)	<0.001
Delay between SRT and cerebral recurrence (months)					0.448
	Mean	10.8 [1–72]	9.2 [1–72]	11.4 [1–70]	
	Median	5	3	5	
Death		173 (87.4%)	132 (86.8%)	41 (89.1%)	0.804
Delay between SRT and death (months)					<0.001
	Mean	12.7 [0–87]	9 [0–87]	24.8 [2–83]	
	Median	7	5.5	19	

## Data Availability

Data are available on request due to privacy restrictions. The data presented in this study are available on request from the corresponding author.

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
