# Peer review of "Initial Age and Performans Status: Predicators for Re-Irradiation Ability in Patients with Relapsed Brain Metastasis after Initial Stereotactic Radiotherapy"

_cancers, 2024, doi:10.3390/cancers16142602_

Round 1

Reviewer 1 Report

Comments and Suggestions for Authors

The submitted manuscript presents the retrospective study, comparing the data of patients who were treated with stereotactic radiation therapy for a single brain metastases (BMs) at the first irradiation and compare characteristics and follow-up outcomes of two subgroups, including one group in which patients were reirradiated for consecutive BMs, while the other was not.

The study was well designed, presented, described and motivated. I think it can be published after addressing several comments, presented below.

Line 69, here, a single sentence should be added what was the expected outcome/benefit of this analysis.

Lines 77, 79, please add percentage values

Table 1, the Authors should decide whether or not use the decimals while presenting the percentage values. I.e, Female, cohort I and II, it is inconsistent now.

Table 2, it should be 0.930 and not 0.93

Figure 1, the y-axis should be limited to 1.0. The same comment applies to Figure 2.

Figure 1, why the Cohort 2 ends after c.a. 105 days?

Line 309, it should also be stated that the patients originated from the single institution

The data presented in Table A could be also shown in a form of a figure, to facilitate the analysis.

Reviewer 2 Report

Comments and Suggestions for Authors

This study compared the efficacy of stereotactic radiation therapy for patients with recurrent brain tumors.    This retrospective study tried to understand why some patients, initially treated for a single metastatic lesion had a longer survival allowing them to be eligible for a second radiation treatment session while others could not reach a second session of radiation therapy. Approximately 50 to 70% of patients who had received initial radiation therapy developed a new metastatic lesion within one year after the first radiation therapy treatment.  Some patients initially treated for a single metastatic lesion had a longer survival allowing them to be eligible for a second radiation treatment session while others could not reach a second session of radiation therapy.  All the neurological symptoms reported by the patients before irradiation remained stable throughout the radiation therapy session.  The findings suggest that the combined score, based on age and Karnofsky score can serve as a predictive factor for improved survival in this study population.  I do not appreciate much that is new in this study.

Comments on the Quality of English Language

English needs minor revisions.

Round 2

Reviewer 2 Report

Comments and Suggestions for Authors

This study compared the efficacy of stereotactic radiation therapy for patients with recurrent brain tumors.    This retrospective study tried to understand why some patients, initially treated for a single metastatic lesion had a longer survival allowing them to be eligible for a second radiation treatment session while others could not reach a second session of radiation therapy. Approximately 50 to 70% of patients who had received initial radiation therapy developed a new metastatic lesion within one year after the first radiation therapy treatment.  Some patients initially treated for a single metastatic lesion had a longer survival allowing them to be eligible for a second radiation treatment session while others could not reach a second session of radiation therapy.  All the neurological symptoms reported by the patients before irradiation remained stable throughout the radiation therapy session.  The findings suggest that the combined score, based on age and Karnofsky score can serve as a predictive factor for improved survival in this study population.  

Comments on the Quality of English Language

English is reasonably good.